# What Should I do and Who's to blame? A cross-national study on youth's attitudes and beliefs in times of COVID-19

Elisabeth L. De Moor[1], Ting-Yu Cheng[1], Jenna E. Spitzer[2], Christian Berger[3], Alexia Carrizales[4], Claire F. Garandeau[5], Maria Gerbino[6], Skyler T. Hawk[7], Goda Kaniušonytė[8], Asiye Kumru[9], Elisabeth Malonda[10], Anna Rovella[11], Yuh-Ling Shen[12], Laura K. Taylor[13], Maarten van Zalk[14], Susan Branje[1], Gustavo Carlo[15], Laura Padilla Walker[16], Jolien Van der Graaff[1] *

1 Department of Youth and Family, Utrecht University, Utrecht, The Netherlands, 2 Department of Developmental Psychology, Utrecht University, Utrecht, The Netherlands, 3 School of Psychology, Pontificia Universidad Católica de Chile, Santiago, Chile, 4 Department of Human Development and Family Studies, Purdue University, West Lafayette, IN, United States of America, 5 INVEST Flagship, University of Turku, Turku, Finland, 6 Department of Psychology, Sapienza University of Rome, Rome, Italy, 7 Educational Psychology, Chinese University of Hong Kong, Hong Kong, China, 8 Institute of Psychology, Mykolas Romeris University, Vilnius, Lithuania, 9 Department of Psychology, Ozyegin University, Istanbul, Turkey, 10 Department of Basic Psychology, University of Valencia, Valencia, Spain, 11 Psychology Department, San Luis National University, San Luis, Argentina, 12 Department of Psychology, National Chung Cheng University, Chiayi, Taiwan, 13 School of Psychology, University College Dublin, Dublin, Ireland, 14 Developmental Psychology, Osnabrück University, Osnabrück, Germany, 15 School of Education, University of California, Los Angeles, Los Angeles, California, United States of America, 16 School of Family Life, Brigham Young University, Provo, UT, United States of America

* j.vandergraaff@uu.nl

**Data Availability Statement:** The anonymized data is made available on the project OSF page: https://osf.io/738z5/. If researchers want to use the data,

## Abstract

The COVID-19 crisis has had a major impact on youth. This study examined factors associated with youth's attitudes towards their government's response to the pandemic and their blaming of individuals from certain risk groups, ethnic backgrounds, and countries or regions. In a sample of 5,682 young adults ($M_{age}$ = 22) from 14 countries, lower perceived burden due to COVID-19, more collectivistic and less individualistic values, and more empathy were associated with more positive attitudes towards the government and less blaming of individuals of certain groups. Youth's social identification with others in the pandemic mediated these associations in the same direction, apart from the COVID-19 burden on attitudes, which had a positive indirect effect. No evidence of country-level moderation was found.

## Introduction

The COVID-19 crisis has had a major impact on the lives of youth across the globe. Although their physical health risks from the virus were relatively low, young adults experienced disproportionally high social and financial costs due to the restrictions imposed by governments to curb the spread of the virus [1], as well as costs for their mental health [2].

they are asked to submit a data request to the project executive team (Jolien Van der Graaff, Laura Padilla-Walker, Gustavo Carlo, Susan Branje). If a project proposes to use data from a single country, approval must be obtained from the lead PIs from that country. If the project requires data from multiple countries, the proposed research needs to be approved by the lead PIs in those countries, as well as the project executive team. A full list of project contributors can be found at: https://youngincoronatimes.sites.uu.nl/international-collaborators/.

**Funding:** JvdG received a grant from the Faculty of Social Sciences of Utrecht University to fund data collection (no grant number available). URL: https://www.uu.nl/en/organisation/faculty-of-social-and-behavioural-sciences ELdM and SB were supported by a grant from the European Research Council (ERC-CoG INTRANSITION-773023). URL: https://erc.europa.eu/ The funders had no role in study design, data collection and analysis, decision to publish, or preparation of the manuscript.

**Competing interests:** The authors have declared that no competing interests exist.

Yet, despite these expressions of negative attitudes towards governments and others, collective crises like the COVID-19 pandemic are also known to increase positive attitudes [3]. Individuals can experience collective identity based on their membership in certain groups [4, 5], and emergencies sometimes create new groups to identify with; that is, all others who are affected by the same adversity [3–6]. These feelings of shared social identity among those affected may result in positive and supportive attitudes towards other individuals and the government on the one hand. On the other hand, they may result in an increased sense of "we versus them" [4, 5], and as such increase the tendency to blame others for experienced adversities. Personal experiences during a crisis (i.e., the perceived burden), as well as personal characteristics (i.e., value orientation and empathy) are thought to play a role in individuals' tendency to identify with others [7–11], and as such may link directly to youth's attitudes towards the government and other citizens during the pandemic, but also indirectly via social identification.

In the present study, we examined among young adults from 14 countries 1) their attitudes towards their government's approach to handling the pandemic and 2) the extent to which they blamed individuals from certain risk groups, ethnic backgrounds, and countries or regions for their negative experiences during the pandemic, which reflect their attitudes towards the institutions and individuals in a society and may also predict their behavior. COVID-19-related and more general individual factors were examined to explain individual differences in youth's attitudes. In particular, we studied whether youth's perceived burden from the pandemic, their collectivistic and individualistic values, and their empathic tendencies predicted their attitudes. We further investigated whether youth's social identification with others during the pandemic could (partly) explain these associations. Finally, even though previous research has shown links between person characteristics and youth's prosocial attitudes to be consistent across cultures (e.g. [12]), it is important to take differences in country-level contextual factors into account. For instance, the number of COVID-19 cases varied greatly across countries at the time of this study (e.g., very low in Taiwan and high in Chile), and the approach that governments took in response to the pandemic also varied. As a result, citizens experienced many more restrictions in one country (e.g., Argentina) than they did in another (e.g., Finland). Therefore, we explored differences across countries in the strength of predictors based on cumulative COVID-19 cases, individualistic orientation of the country, and the strictness of the national COVID-19 approach at the time of assessment.

## Perceived burden as a consequence of the COVID-19 pandemic

Individual differences between youth in the extent to which they perceived a burden due to the COVID-19 pandemic may help understand differences in their attitudes and beliefs. Individuals living through the pandemic may have faced a myriad of changes in their social, medical, and financial situation [1], either as a direct consequence of the virus (e.g., becoming sick yourself or seeing others get sick), or as a consequence of the restrictions put in place to regulate the spread of the virus (e.g., closing of businesses resulting in lay-offs, social distancing resulting in less social contact). These restrictions may have been especially impactful for this age group, as young adults are more likely to live alone than mid- and late-adults and thus to become isolated [13], and they tend to be in a less secure job situation [14, 15], which may have contributed to a sense of distress [15].

The personal experience of the COVID-19 pandemic may have affected youth's attitudes towards the government and others in the society, as they may have shifted blame to the government and to other citizens to alleviate their own stress and relieve responsibility (Murray, [unpublished]). In particular, those individuals who perceived the burden to be high may have had more negative *attitudes towards the government approach*, given the role of the

government in putting in place restrictions that were part of the burden that they experienced or that were ineffective at decreasing it. For instance, individuals from the UK who felt that the satisfaction of their need for connectedness was thwarted by the restrictive measures, and thus perceived a greater burden, reported lower trust in their government [16]. Also, youth who perceived a high burden may also have been more likely to *blame certain groups* for their negative experiences [17]. Experiencing stress may deplete individuals' cognitive and emotional resources that are necessary to attend to another's situation and to respond in a prosocial way (see [18]). Indeed, individuals who had contracted COVID-19 and who, as a result, experienced a lot of distress were found to blame others for their non-supportiveness during their suffering, such as doctors, neighbors, and the community at large [19]. Thus, we expected youth who perceived a higher COVID-19-related burden to be less positive about the government's approach and to be more likely to blame certain other groups for their distress.

## Collectivistic and individualistic values

Youth's collectivistic and individualistic values may also relate to their attitudes towards the government and other groups in society. Although the individualism-collectivism spectrum has traditionally been used to classify and contrast countries [20], it is important to note that within countries, individuals also differ in the extent to which they hold individualistic (i.e., distinctiveness from others, high self-reliance, and self-promotion) and collectivistic (i.e., similarity to others, interdependence, community-promotion) values. Individuals who hold more individualistic values are less supportive of government interventions on national welfare, state ownership, and market competition [21]. In contrast, individuals with more collectivistic values tend to feel more responsibility towards, and greater identification with, their community [22–24]. They also tend to show greater compliance with, and adherence to, social norms set forth by that community on average [25]. In the context of the COVID-19 pandemic, individuals with more collectivistic values have indeed been found to be more inclined to comply with the cautionary measures [26, 27]. Therefore, youth who held more collectivistic and less individualistic values can be expected to have had more positive *attitudes towards the government approach* to combat the pandemic.

Regarding attitudes toward other groups, however, having more collectivistic and less individualistic values may be related with more *blaming of certain groups* that are not considered part of one's community [28]. Having higher collectivistic values has indeed been associated with a stronger differentiation between in- and outgroup [29]. In line with this finding, during the COVID-19 pandemic youth with collectivistic values tended to be more afraid and worried than youth with less collectivistic values of being infected by others, such as foreigners (outgroup), or something unknown, like a novel virus [27]. Youth with higher levels of collectivistic values and less individualistic values might thus have more negative attitudes towards–and even blame–groups in the society that they see as responsible for the spread of COVID-19 or for the restrictive measures.

## Empathy

Youth's empathy may also be associated with their attitudes towards the government and towards others during the COVID-19 pandemic. Although definitions of empathy vary, empathy can be viewed as a person's responses to other's experience, including both cognitive processes like understanding other's perspective and emotional processes like feeling concern for others [30]. As highly empathic individuals are generally sensitive to the needs of others and are understanding of others' situation, they tend to show prosocial attitudes and behaviors [31]. Higher empathy has indeed been related to a wide range of prosocial tendencies (e.g., [32, 33]).

Indeed, in the context of the pandemic, levels of empathy may explain differences in youth's attitudes and blaming. Regarding *attitudes towards the government approach*, more empathic individuals have been found to be more positive towards government intervention [34] and to hold more social and liberal political views in general [35]. Thus, youth with higher levels of empathy may have also held more positive views regarding the national approach to combatting COVID-19. With regard to *blaming of certain groups*, higher levels of empathy have also been associated, developmentally and experimentally, to lower prejudice and lower bias towards outgroups (e.g., [36–39]). During the COVID-19 pandemic, youth who were more empathic may therefore have had a lower tendency to blame certain groups.

## The mediating role of social identification

Youth's social identification with others who were affected by the COVID-19 pandemic may in part mediate the above-mentioned pathways. That is, youth who identified with other individuals who were affected may have shown greater understanding for restrictions imposed by the government and have been less likely to blame others. This is in line with work showing that greater identification with the community predicted more understanding and prosociality during previous crises (e.g., [3, 40]) and during the COVID-19 pandemic (i.e., providing support and adherence to regulations; [41]). Individuals who feel more connected to others and, as such, experience a greater sense of "we-ness" or identification with their ingroup may more highly value the benefits to be reaped by society.

Both collectivism and empathy have been related to social identification [8–10], with collectivistic and empathic individuals generally having a stronger sense of social identification. Furthermore, in line with the notion of "*altruism born of suffering*" (e.g., [42, 43]), we may expect that youth who experienced a greater COVID-19-related burden identified more with others that were affected by the pandemic [7, 11]. Thus, we expected that youth who had higher levels of empathy, collectivism and COVID-19-related burden, and lower levels of individualism experienced greater social identification with those affected by the COVID-19 pandemic and, in turn, exhibited more positive attitudes towards the government approach and less blaming of certain groups.

## The current study

Following the idea that the COVID-19 crisis may have impacted youth's relations with others and their social identities, the present study investigated whether perceived COVID-19 burden, collectivistic and individualistic values, and empathy could help us understand individual differences in youth's attitudes and beliefs during the COVID-19 pandemic. We examined social identification as a mediator and explored moderation at the country level to uncover potential cross-country differences in these associations. Specifically, we examined moderation by the cumulative COVID-19 cases per 100,000 inhabitants, the strictness of the national approach as factors of COVID-19-related country differences, and the national-level individualistic orientation as a more general country difference factor, as the association between individual-level factors may depend on both the specific COVID-19 situation and general values in a country (see Fig 1 for a complete conceptual overview of the tested associations).

We expected that youth who reported a greater perceived burden due to COVID-19 would report less positive attitudes towards the government approach and would be more likely to blame others for their distress. Youth who held more collectivistic and less individualistic values were expected to report more positive attitudes towards the government approach and more blaming of groups they perceive as an outgroup. We also expected youth with greater empathy to express more positive attitudes towards the government approach and less blaming

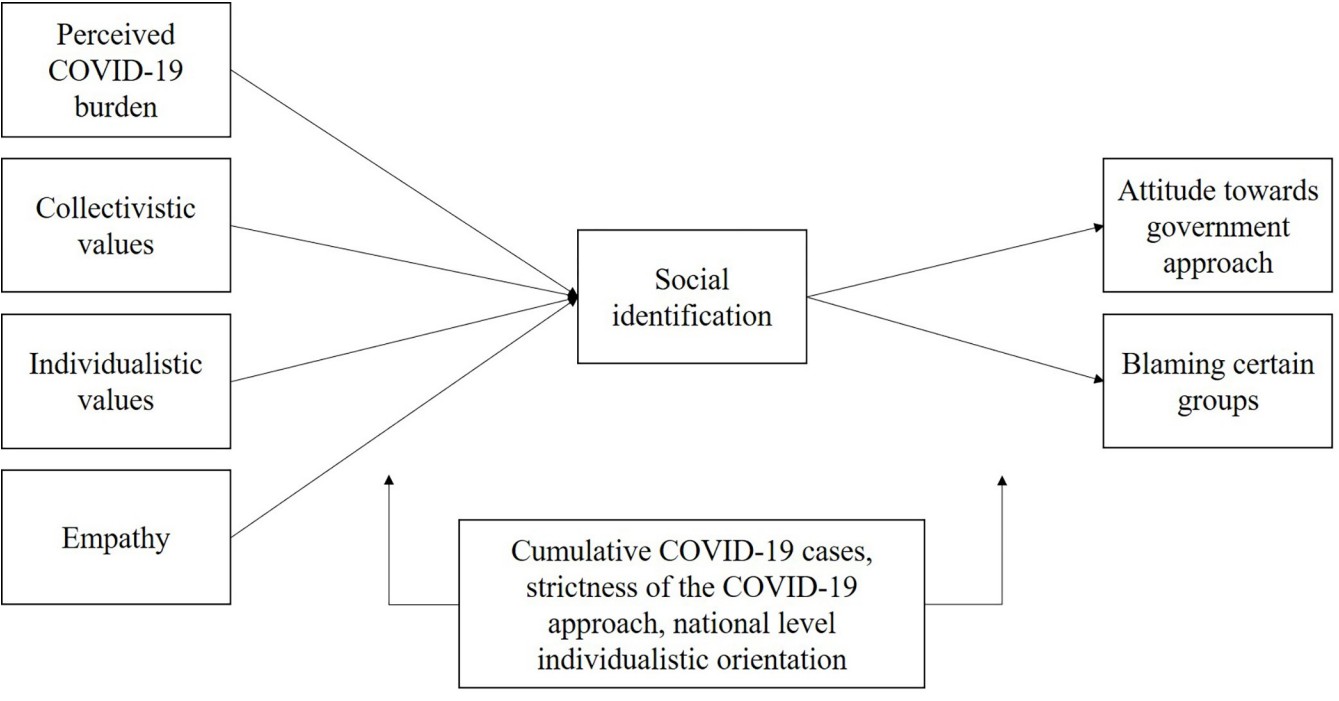

**Fig 1. Conceptual model of the studied relations.**

of certain groups. Furthermore, we hypothesized indirect effects via social identification with others during the COVID-19 pandemic in the same direction for collectivistic and individualistic values and empathy. For perceived burden, we expected a higher burden to be associated with higher social identification and thus indirectly associated with more positive attitudes and less blaming of certain groups. Finally, we did not form any specific hypotheses regarding country-level moderation. The present study's research questions, hypotheses, methods, and analyses were pre-registered at https://osf.io/et2nb.

## Materials and methods

### Participants and procedures

The faculty ethics review board of the Faculty of Social and Behavioral Sciences of Utrecht University approved the study (20–298). Informed consent was obtained in written form from all participants. Participants were 5,682 young adults aged 18–25 years (67.1% female; $M_{age}$ = 21.51 years, $SD_{age}$ = 2.13) from 14 countries: Argentina ($n$ = 253), Chile ($n$ = 146), China ($n$ = 286), Finland ($n$ = 465), France ($n$ = 506), Germany ($n$ = 375), Ireland ($n$ = 495), Italy ($n$ = 502), Lithuania ($n$ = 346), Spain ($n$ = 179), Taiwan ($n$ = 169), the Netherlands ($n$ = 472), Turkey ($n$ = 372), and the United States ($n$ = 1,116). Participants were mostly of medium to high socioeconomic status (SES), with 76.1% reporting some form of postsecondary education (23.8% reported secondary education, 1 participant reported primary education) and 38.0% indicating to live comfortably on the present household income (44.9% coping on present income, 13.8% having some difficulties with the present income, 3.3% having many difficulties on the present income). These youth participated in #Young in Times of COVID-19, a study evaluating prosocial behavior during the COVID-19 pandemic among youth. Participants were primarily recruited via the networks of faculty collaborators in each country. However,

participants from the Netherlands and the United States were also recruited using survey panel methods. Respondents received no or minimal (no more than 5 euros) compensation. Participants completed an online survey throughout the summer and fall of 2020 in the primary language of their county. For several of the questionnaires included, versions in the primary languages were already available; questionnaires for which no primary language version existed were translated by collaborators from each of these countries. The survey took approximately 20 minutes to complete. Only participants who completed the survey and passed the validation checks (e.g., "If you are reading this, select 'a lot'") were included in the final dataset. See S1 Table for participants' demographic information by country.

## Measures

**Outcome variables.** *Attitude towards government approach*. Participants reported on their attitudes towards their country's government approach to COVID-19 using a 3-item scale developed for this research. Participants expressed the extent to which they agreed that the government in their country "takes effective measures to combat COVID-19", "protects all citizens against the negative effects of COVID-19", and "explains its decisions about the approach to combat COVID-19". Responses were rated on a Likert scale of 1 (*strongly disagree*) to 5 (*strongly agree*) with higher scores reflecting a more positive attitude towards their government approach. Importantly, these items were not included in the Chinese questionnaire. The scale exhibited good internal consistency (Cronbach's α = .84).

*Blaming of certain groups for COVID-19*. Participants reported on their negative outgroup beliefs using 4 items that were created for the current study. Respondents answered the following question, "To what extent do you blame the following groups for the negative consequences you experience from the COVID-19 pandemic?" for the following four groups: "Citizens from certain ethnic groups", "Citizens who do not follow the recommendations to prevent the spread of the virus", "Citizens who belong to so-called risk groups (e.g., elderly, people with pre-existing health conditions)", and "Citizens of certain countries or regions". Responses were rated on a Likert scale from 1 (*do not blame them at all*) to 4 (*blame them completely*), with higher values reflecting greater blame. The complete scale exhibited questionable internal consistency (Cronbach's α = .57). Therefore, and following our pre-registered plan, we removed one item, which pertained to the second item described above. This led to an acceptable internal consistency of Cronbach's α = .69. The three-item scale was used in further analyses. To test for robustness, we also ran our analyses using the original four-item scale. Those findings did not meaningfully differ from the findings reported in the manuscript. After excluding the worst-fitting blaming item, however, the effect of the perceived burden on blaming was somewhat weaker and the effect of empathy on blaming somewhat stronger, although both effects were significant in the analyses with and without the worst-fitting item.

**Predictors.** *Perceived COVID-19 burden*. Participants reported on the extent to which they were burdened by the COVID-19 pandemic using 9 items composed specifically for this study (see S2 Table). The scale covered several aspects of burden, including health (2 items; e.g., "being or become infected with or having symptoms of COVID-19"), social situation (4 items; e.g., "being restricted in visiting friends or family"), and finance (3 items; e.g., "being at an increased risk for lower financial means"). Participants rated the items on a Likert scale of 1 (*not at all burdensome*) to 5 (*very burdensome*), with higher scores reflecting greater perceived burden. The scale exhibited good internal consistency (Cronbach's α = .76).

*Collectivistic and individualistic values*. Orientations towards collectivism and individualism were measured using two subscales (with 8 items each) from the validated Culture Orientation Scale [44]. Respondents rated the extent to which various statements about culture

orientation reflected their values using a Likert scale from 1 (*strongly disagree*) to 7 (*strongly agree*), with higher scores reflecting greater individualism or greater collectivism. Sample items include, "If a coworker gets a prize, I would feel proud" (collectivistic values), and "I often do 'my own thing'" (individualistic values). The scale exhibited good internal consistency (Cronbach's α = .70 for individualistic values, .72 for collectivistic values).

*Empathy*. Participants reported on two empathy-related constructs, namely, empathetic concern and perspective taking, using 10 items adapted from the Interpersonal Reactivity Index (IRI; [30]). Responses were based on a 5-point Likert scale ranging from 1 (*does not describe me at all*) to 5 (*describes me greatly*). Sample items include, "I would describe myself as a pretty soft-hearted person" and, "I sometimes find it difficult to see things from the other person's point of view". Items were reverse coded when necessary, such that higher scores indicated more empathy. Scores from both subscales were averaged to create one empathy score. The empathy scale exhibited good internal consistency (Cronbach's α = .75), and the IRI has procured ample evidence of psychometric validity and reliability (see [45] for an overview).

*Social identification with others during COVID-19*. Participants reported on their social identification with others during the COVID-19 pandemic using 4 items adapted from Doosje et al. [46]. On a Likert scale of 1 (*strongly disagree*) to 5 (*strongly agree*), participants responded to items such as "During the COVID-19 crisis. . . I identified with the other people affected". Higher scores reflect greater social identification, and the scale exhibited good internal consistency (Cronbach's α = .78).

**Country-level moderators.** *Cumulative cases of COVID-19 per 100,000 inhabitants*. The cumulative number of COVID-19 cases, collected from the World Health Organization's official Coronavirus Dashboard [47], was used to characterize the extent of the outbreak in each country at the end of data collection. For Taiwan, data were collected from the Taiwan Centers for Disease Control official website [48]. To make meaningful cross-national comparisons, we computed the number of cumulative cases per 100,000 inhabitants, based on the population numbers of the United Nations Population Division estimates [49].

*Strictness of COVID-19 approach*. We used the Stringency Index of the Oxford COVID-19 Government Response Tracker to characterize the level of action each government undertook in response to the pandemic [50]. This measure assigns a level of strictness on a scale from 0 to 100 based on aggregated indicators of government responses, such as school closures and travel restrictions, plotted against the coronavirus cases and deaths in each country. We computed the strictness of each government's response by averaging the daily strictness ratings for each country from the start to the end of data collection in each country.

*National level individualistic orientation*. We assessed individualistic values at the national level using the Hofstede's Index [51]. The individualism scale reflects the extent to which each society prioritizes the pursuit of individual goals over collective goals. Higher ratings indicate a stronger emphasis on individualistic values.

## Statistical plan

Prior to registering our study, a power analysis was conducted to estimate the required sample size to detect effects in the mediation model and the extended moderated mediation model. With our complete sample of 5,682 participants, a power of .80, and significance at α = .05, effects of $f^2$ = .002 (very small effect size) could be detected for the mediation model. In the country-level multigroup models which had subgroup sizes of at least 1,116 participants (for the "high" group of national-level individualistic orientation), effects of at least f2 = .011 (small effect size) could be detected.

**Direct effects and mediation models.** To test our research questions, we tested a series of models in Mplus [52]. The analysis scripts and anonymized data are made available on the project OSF page: https://osf.io/738z5/. We first estimated a model with four predictors (i.e., perceived burden, collectivistic values, individualistic values, and empathy) and two outcomes (i.e., attitude towards government approach and blaming of certain groups). This model was estimated for the participants from all countries together. Then, we added social identification as a mediator of all predictive pathways from independent to dependent variables to the model. An alpha of .05 was used to determine significance of effects. Effect size was evaluated using rules of thumb by Funder & Ozer [53] and the $R^2$ statistic.

**Country-level moderation models.** Next, to examine moderation by country-level variables, we divided the sample into groups of low, medium, and high on each of the country-level factors. As preregistered, this was done because the number of countries was too small to test a multilevel mediation model, for which a minimum number of 50 clusters is suggested when used in a Structural Equation Framework with Maximum Likelihood (ML) estimation [54, 55]. For cumulative cases per 100,000 inhabitants and for the strictness of the COVID-19 approach, we used percentile scores to divide the countries into three categories because there were, to the best of our knowledge, no informative grounds at the time to base this classification on in the broader literature. We separated the countries into three categories of "low" (1st to 33th percentile), "medium" (34th to 66th percentile), and "high" (67th to 100th percentile). For national level individualistic orientation, we used the cutoffs described for this Hofstede dimension on the Hofstede Insights website [51, 56]. These cutoffs describe "low" (below 50), "high" (above 60), and "very high" values (above 80). For the present study, we adapted this scale to cover all possible values, such that "low" was 0–50, "medium" was 51–80, and "high" was 81–100.

Then, to examine moderation by these country-level variables, we estimated a multigroup model for each of the country-level factors. The three scores on the country-level variables (i.e., low, medium, and high) were used to divide the different countries into three groups. We then tested a model for each country-level variable in which the predictive paths were constrained to be equal across groups. If model fit was appropriate, we released groups of predictive parameters in steps, to see whether allowing the predictive paths to vary between groups would significantly improve the fit to the data. Originally, we planned to test two models per country-level variable: one with the predictive pathways constrained and one fully unconstrained, and to compare their model fit. However, the unconstrained model was fully saturated and therefore no model fit statistics were available to be compared with the constrained model. Instead parameters were released per outcome variable, so that we tested three partially unconstrained models: one with unconstrained pathways towards attitudes towards the government approach, one with unconstrained pathways towards blaming of certain groups, and one with unconstrained pathways towards social identification. If one of the partially unconstrained models fitted significantly better than the constrained one, this was taken as evidence for country-level moderation. If this was the case, we further examined for this model on which predictive pathways the differences existed by releasing the constraints of the individual paths one at a time. Model fit of CFA $\geq$ .900 and RMSEA of $\leq$ .080 was deemed acceptable [57, 58]. Change of $\Delta$CFI $\geq$ .005, supplemented by $\Delta$RMSEA $\geq$ -.010 was seen as indicative of significant change in model fit [59], indicating that the (partially) unconstrained model fit better than the constrained model.

## Results

### Descriptive statistics

Descriptive statistics of, and correlations between, the study variables are provided in Table 1, respectively (see S1 Table for a breakdown of the descriptive statistics per country). On

**Table 1. Descriptive statistics of and correlations between the study variables.**

| | 1. | 2. | 3. | 4. | 5. | 6. | 7. |
|---|---|---|---|---|---|---|---|
| 1. COVID-19 burden | | | | | | | |
| 2. Individualistic values | .14* | | | | | | |
| 3. Collectivistic values | .25* | .09* | | | | | |
| 4. Empathy | .20* | -.07* | .40* | | | | |
| 5. Social identification | .20* | .02 | .36* | .29* | | | |
| 6. Attitude towards government approach | -.05* | -.01 | .11* | .08* | .19* | | |
| 7. Blaming of certain groups | .13* | .21* | -.02 | -.20* | .04* | .01 | |
| Mean | 3.16 | 4.54 | 5.18 | 3.83 | 3.18 | 3.43 | 1.37 |
| SD | 0.79 | 0.87 | 0.82 | 0.57 | 0.92 | 1.06 | 0.55 |

average, participants scored above the middle of the scale on all variables, except blaming of certain groups, which was reported comparatively little. Furthermore, most correlations were significant, with effects ranging from small to medium in size. Most of the correlations were in the expected direction, although notably, higher social identification was associated with more blaming of certain groups. Moreover, there were no significant correlations between having individualistic values and social identification, and between individualistic values and the attitude towards the government approach. Moreover, having collectivistic values was not significantly associated with the blaming of certain groups.

## Main results

**Direct effects model.** We first tested a main effects model (Table 2), which was fully saturated and thus yielded no interpretable model fit statistics (i.e., CFI = 1.000, RMSEA = .000). Perceiving a greater burden due to COVID-19 was related to a less positive attitude, whereas having more collectivistic values and being more empathic was related to a more positive attitude towards the government approach. Having more individualistic values was not significantly related to youth's attitudes. Perceiving a greater COVID-19 burden and having stronger individualistic values were associated with more blaming of certain groups, whereas higher

**Table 2. Direct effects regression model with attitude towards government approach and blaming of certain groups regressed on COVID-19 burden, individualistic values, collectivistic values, and empathy.**

| | b | SE(β) | β | p |
|---|---|---|---|---|
| **Predictors of attitude towards government approach** | | | | |
| COVID-19 burden | -0.12 | .01 | -0.09 | < .001 |
| Individualistic values | -0.01 | .01 | -0.01 | .587 |
| Collectivistic values | 0.14 | .02 | 0.11 | < .001 |
| Empathy | 0.10 | .02 | 0.05 | < .001 |
| **Predictors of blaming of certain groups** | | | | |
| COVID-19 burden | 0.11 | .01 | 0.15 | < .001 |
| Individualistic values | 0.11 | .01 | 0.17 | < .001 |
| Collectivistic values | 0.01 | .01 | 0.02 | .288 |
| Empathy | -0.21 | .01 | -0.22 | < .001 |
| **Covariances** | | | | |
| Attitude towards government approach × Blaming of certain groups | 0.02 | .01 | 0.03 | .011 |

Note. $R^2$ for attitude towards government approach = 2.0%

$R^2$ for blaming of certain groups = 9.8%

**Table 3. Mediation model with attitude towards government approach and blaming of certain groups regressed on COVID-19 burden, individualistic values, collectivistic values, and empathy, mediated by social identification.**

| | Direct effects | | | | Indirect effects | | | |
|---|---|---|---|---|---|---|---|---|
| | b | SE(β) | β | p | b | SE(β) | β | p |
| **Predictors of attitude towards government approach** | | | | | | | | |
| COVID-19 burden | -0.14 | .01 | -0.11 | < .001 | 0.03 | < .01 | 0.02 | < .001 |
| Individualistic values | < -0.01 | .01 | < -0.01 | .970 | < -0.01 | < .01 | < -0.01 | .575 |
| Collectivistic values | 0.08 | .02 | 0.06 | < .001 | 0.07 | .01 | 0.05 | < .001 |
| Empathy | 0.04 | .02 | 0.02 | .142 | 0.05 | < .01 | 0.03 | < .001 |
| Social identification | 0.21 | .01 | 0.19 | < .001 | – | – | – | – |
| **Predictors of blaming of certain groups** | | | | | | | | |
| COVID-19 burden | 0.10 | .01 | 0.14 | < .001 | 0.01 | < .01 | 0.01 | < .001 |
| Individualistic values | 0.11 | .01 | 0.17 | < .001 | < 0.01 | < .01 | < -0.01 | .576 |
| Collectivistic values | < 0.01 | .02 | -0.01 | .731 | 0.01 | < .01 | 0.02 | < .001 |
| Empathy | -0.22 | .01 | -0.23 | < .001 | 0.01 | < .01 | 0.01 | < .001 |
| Social identification | 0.04 | .01 | 0.07 | < .001 | – | – | – | – |
| **Predictors of social identification** | | | | | | | | |
| COVID-19 burden | 0.12 | .01 | 0.10 | < .001 | – | – | – | – |
| Individualistic values | -0.01 | .01 | -0.01 | .574 | – | – | – | – |
| Collectivistic values | 0.31 | .01 | 0.27 | < .001 | – | – | – | – |
| Empathy | 0.25 | .01 | 0.16 | < .001 | – | – | – | – |
| **Covariances** | | | | | | | | |
| Attitude towards government approach × Blaming of certain groups | 0.01 | .01 | 0.03 | .071 | – | – | – | – |

*Note.* $R^2$ for attitude towards government approach = 4.9%, $R^2$ for blaming of certain groups = 16.5%, $R^2$ for social identification = 10.2%

empathy was related to less blaming. Having collectivistic values was not associated with youth's blaming of certain groups. All significant regression coefficients fell in the range of relatively small effects (around $\beta$ = .10; [53]). Total explained variance in the dependent variables was also limited (Table 2).

**Indirect effects model.** Next, we included social identification in the model as a mediating variable (Table 3). Again, the resulting model was fully saturated, and did not allow us to gauge model fit. The direct effects of COVID-19 burden, individualistic values, collectivistic values, and empathy did not differ meaningfully from those in the direct effects model. In addition, social identification was also a significant predictor of a more positive attitude towards the government approach and, surprisingly, of more blaming of certain groups. Moreover, youth's perceived COVID-19 burden, collectivistic values, and empathy were significantly related to more social identification. Individualistic values were not significantly associated with social identification. As for the direct effects model, most effect sizes were relatively small.

Regarding indirect effects, we found a positive indirect effect of empathy on attitude towards the government approach via social identification, which was in line with the direct effect. We also found a positive indirect for perceived burden, whereas the direct effect was negative. This indicates that in addition to perceived COVID-19 burden being directly associated with a less positive attitude towards the government approach, higher perceived burden was also associated with more social identification, which in turn was associated with a more positive attitude. Youth's perceived COVID-19 burden, their collectivistic values, and their empathy were also indirectly related to blaming of certain groups via social identification. The indirect effect of COVID-19 burden was in the same (positive) direction as the direct effects,

but the indirect effect of empathy was not. Specifically, although youth's empathy was directly related to lower blaming, indirectly it was related to more blaming due to the positive association of social identification with blaming of certain groups. For collectivistic values, there was no direct association with blaming of certain groups, but there was a positive indirect association via social identification. All significant indirect effects were very small in size (around $\beta$ = .05; [53]), and total explained variance of the dependent variables was limited (Table 3).

**Country-level moderation models.** To examine moderation by country-level variables, we fitted multigroup models using each of the country-level variables as a grouping variable to group participants of the 14 countries into a low, medium, or high group. We first fitted the model using *cumulative COVID-19 cases* as a grouping variable, constraining all predictive paths including indirect paths to be equal across groups. This model fit the data well (CFI = .932, RMSEA = .051; see S3 Table for the predictive pathway parameters of this model). To examine whether a partially unconstrained model would fit the data better, we then tested three subsequent models: one in which all direct paths on youth's attitudes towards the national approach were allowed to vary between groups, one in which all direct paths on blaming of certain groups was allowed to vary, and one in which all direct paths on social identification were allowed to vary. All of the partially unconstrained models fit the data well (CFI = .959, RMSEA = .050; CFI = .961, RMSEA = .049; CFI = .946, RMSEA = .055, respectively), but did not significantly improve model fit.

Then, we examined the *strictness of the COVID-19 approach* as a potential moderator by fitting a multigroup model in which all predictive pathways were constrained to be equal across strictness groups. This model fit the data acceptably (CFI = .932, RMSEA = .052; see S4 Table for the predictive parameters). We once more tested three partially unconstrained models, with predictive pathways released for attitudes towards national approach, blaming of certain groups, and social identification. All of these models fit the data well (CFI = .947, RMSEA = .057; CFI = .964, RMSEA = .047; CFI = .951, RMSEA = .052, respectively), but none significantly improved the fit of the constrained model.

Finally, we explored *national level individualistic orientation* as a moderator, again splitting our sample into three groups of low, middle, and high, and comparing these groups in a multigroup model. The constrained model fit the data well (CFI = .943, RMSEA = .045; see S5 Table for the predictive parameters). Models with all prospective pathways unconstrained on attitudes towards the government approach, blaming of certain groups, and social identification also fit the data well (CFI = .969, RMSEA = .041; CFI = .965, RMSEA = .044; CFI = .951, RMSEA = .049, respectively), but did not significantly improve model fit. Thus, we concluded that country-level differences in the cumulative COVID-19 cases, the strictness of the national approach, and the national level individualistic orientation did not significantly moderate the examined associations.

## Discussion

The COVID-19 pandemic and the restrictions put in place to combat it have had a great impact on young adults' daily lives, and may have impacted the ways in which youth relate to others in society [6]. The present study aimed to help us explain differences in youth's attitudes towards their national government's approach to combatting the virus and their tendency to blame individuals in risk groups or from certain ethnic backgrounds, countries, or regions. Our findings show that youth's perceived burden due to COVID-19, collectivistic and individualistic values, and empathy indeed played a role. The extent to which youth identified with other people during the pandemic mediated these associations, which was consistent across different countries.

## Perceived burden as a consequence of the COVID-19 pandemic

In line with our expectations, youth who perceived a greater burden had, on average, more negative *attitudes towards the national approach* to combatting the spread of the virus. This result is in support of the notion that experiencing stress depletes individuals' cognitive and emotional resources that are necessary to attend to others' needs [18]. As such, youth who perceived high burden during the COVID-19 pandemic may have been less able to attend to the consequences of the pandemic for other. As a result, they may have been less supportive of the government's approach to handle these collective consequences, in particular when the costs for themselves were high. Another possibility is that they reported a more negative attitude towards the government's approach because they shifted blame for their burden to the government in an attempt to alleviate COVID-19 related distress (Murray, [unpublished]). Our result is in line with previous research that also showed that higher perceived burden was related to more negative attitudes toward the government [16].

Further, in support of our hypothesis, youth who experienced a higher burden were also more likely to *blame certain groups* for the origin or spread of the disease and for the restrictions. Similar processes as for the link with negative attitudes towards the government have likely played a role here. That is, blaming others may have been used by youth as a way of coping with their own distress (Murray, [unpublished]), and the depletion of cognitive and emotional resources due to the burden they experienced may have diminished their tendency to see things from others' point of view [18], which may have been a ground for negative attitudes towards others. It further supports previous research also showing an association between burden as a consequence of COVID-19 and individuals' negative attitudes towards others (e.g., [19]).

Regarding the indirect effects via social identification, perceived burden was also indirectly related to more blaming of certain groups. However, in contrast to the direct effect, perceiving a greater burden as a consequence of the COVID-19 pandemic was indirectly associated with more positive attitudes towards the government approach via social identification. This finding is in line with the notion of "*altruism born of suffering*" [11, 43] and with previous empirical research on the role of stress in prosocial attitudes [7, 42, 60, 61]. This work has showed that experiencing stress can increase one's identification with others who experienced the same stress, and this may facilitate one's motivation to help relieving others' stress. Our results indeed suggest that, if youth's higher perceived burden went together with higher levels of identification with others during the COVID-19 pandemic, this facilitated their understanding and supportiveness of the measures that governments had to take to diminish the consequences of the pandemic for all.

Thus, perceived burden may play both a positive and negative role in youth's attitudes, depending on whether it is paired with a greater sense of identification with the people around them. It is also possible that the way burden is related to one's attitudes depends on the type of burden. For instance, experiencing a burden to one's health (e.g., getting ill) may be tied to greater social identification, whereas experiencing a social burden (e.g., not being able to visit friends) may be tied to less social identification. In the future, it would be interesting to examine the differential effects of these different types of burden on attitudes and behavior. For policy makers, it may be important to keep in mind the negative effect experiencing a burden may have on youth's attitudes towards the government and other members of society, and on their behavior during a pandemic more generally.

## Collectivistic and individualistic values

In line with our hypotheses and earlier work [26, 27], having collectivistic values was directly and indirectly related to more positive *attitudes towards the national approach*. Thus, youth

who held more collectivistic values were more likely to see the benefits of their government's approach to combatting the virus. However, and contrary to our expectations, individualistic values were not related to these attitudes. Although having individualistic values is generally associated with less support for government intervention [21], it is possible that the same does not hold for COVID-19 related interventions. That is, although youth with individualistic values may still not like their government to impose strict rules on them, they may feel personally responsible to contribute to combating this global disease [62], and as such be more supportive of a government approach that is in line with that goal. For future research, it would be helpful to include information on youth's sense of personal responsibility and their opinions regarding what would be the best approach towards handling the COVID-19 pandemic.

In contrast to our hypothesis, our findings showed that holding collectivistic values was not directly related to *blaming of certain groups*. However, holding stronger individualistic values was directly related to more blaming. This might be explained by individuals with individualistic values placing more value on personal responsibility, both for themselves and for others [62]. As such, they may view others as personally responsible for contributing to the pandemic. For instance, they may have considered vulnerable groups responsible for the unwanted government regulations [21]. That holding collectivistic values was not directly related to blaming might be due to the specific groups that were seen by youth as ingroups or outgroups and those that were included in the blaming measure. Whereas some of the items of the measures referred to outgroups (e.g., people from other countries), other items referred to groups that may have been considered ingroups (e.g., people from risk groups). The COVID-19 pandemic may have actually increased identification with these groups [6]. As stronger collectivistic values have been associated with a stronger differentiation between in- and outgroup [29], this may explain how having higher levels of collectivistic values was indirectly related via social identification to more blaming of certain groups. Thus, our findings suggest that creating a sense of "we-ness" rather than a focus on individualistic needs may be important during a global pandemic to foster positive attitudes towards the national approach to combat the spreading of the disease and to limit negative views towards other members of society, but also caution that greater identification with one group may come at the risk of increasing negative views towards other groups. Thus, policy makers during a pandemic should carefully consider the sense of social identity that they promote, and who is and who is not included in this identity.

## Empathy

As expected, higher levels of empathy were directly and indirectly related to more positive *attitudes towards the government approach* to combatting the disease and directly to less *blaming of certain groups*. This suggests that empathy promoted more positive attitudes towards the government and others in society during the COVID-19 pandemic. This is in line with the notion that individuals high in empathy are sensitive to the needs of others and are understanding of others' situations, and as such they are more likely to engage in prosocial attitudes [31]. Moreover, it corroborates findings showing that empathy is associated with a more positive view of government intervention [34] and lower prejudice towards outgroups [36–39, 63, 64]. Our finding that empathy was positively linked to social identification (which in turn was related to more positive attitudes towards the government) is also in accordance with previous work showing that individuals high in empathy have a higher tendency to identify with others (e.g., [65]). These findings suggest that fostering empathy could potentially be an important pathway towards more prosocial attitudes and behaviors during a global pandemic.

## The role of social identification

In addition to the direct and indirect paths described above, social identification was also positively associated with youth's attitudes towards the government approach and blaming of certain groups. Youth with higher levels of social identification reported more positive attitudes towards the government approach but, surprisingly, also more blaming. This might be the case because the individuals youth identified with (e.g., "other people affected") may not overlap fully with the groups one holds responsible for the origin/spread of the disease or the restrictions put in place to combat the disease. Thus, identifying more with individuals may make them feel more strongly for their suffering and, as a result, make them more likely to blame others who they see as responsible for that suffering. Again, in the future it would be helpful to get a better sense of who belongs to young adults' ingroup and outgroup. Regarding mediation effects on blaming of other groups, we found indirect effects for the perceived burden, collectivistic values, and empathy which were opposed to the direct effects and against our expectations. This is probably due to the positive direct association of social identification with blaming described above.

## Country-level moderation

Our exploration of the role of country-level moderators revealed no significant results, suggesting that the studied associations did not differ between countries depending on the cumulative number of COVID-19 cases, the strictness of the national approach, or the national level individualistic orientation. The associations thus seem to be robust across the various countries, which is in line with previous research revealing consistent results on the predictors of prosocial tendencies across different cultures (e.g., [66, 67]). That country-level factors did not moderate the associations in our study can also be explained by the fact that our participants were all in the same developmental stage (i.e., young adulthood) and many of them were college students. According to Jensen's cultural-developmental approach [68], during young adulthood the similarities in individuals' attitudes due to the shared developmental phase, in which all youth strive for increasing independence, are larger than the differences due to variations across cultures and countries. Moreover, due to the COVID-19 pandemic, the challenges that individuals faced were highly similar, which may also have increased similarity across cultures.

Alternatively, although there were no country-level differences depending on the cumulative number of COVID-19 cases, the strictness of the national approach, or the national level of individualistic orientation, it is possible that other factors at the country-level may indicate differences in youth's attitudes towards their government's approach to combatting the spread of the disease and to their blaming of individuals from certain risk groups, ethnic groups, and countries or regions. For instance, the present study focused on individual and country-level differences in collectivistic and individualistic values, but other cultural dimensions may also play a role (e.g., [20, 56]). As a case in point, differences in the uncertainty avoidance dimension (i.e., discomfort with and resistance to unfamiliar phenomena; [56]) have been associated at the country level with spread of the disease and with greater mortality risk [69, 70]. In addition to health outcomes, higher uncertainty avoidance has been related at the country-level to less gathering of people in public places [71] and more consumer stockpiling [72], and at an individual level to greater beliefs in COVID-19 conspiracy theories [73]. Thus, it may be interesting to examine individual- and country-level values also in relation to youth's attitudes towards their government's approach to combatting the spread of the disease and to their blaming of risk groups and individuals from certain ethnic groups and countries or regions.

## Limitations

The present study had some limitations. First, it should be noted that our sample was likely somewhat biased towards high-SES, highly educated individuals due to recruitment having occurred mostly via the researchers' personal networks. Second, the data were cross-sectional. Whereas the present data helped us understand differences between youth in their attitudes towards the government approach and blaming of certain groups, they do not allow us to predict these differences. What is more, we cannot be sure that the directionality examined in this study reflects reality, and it could be that instead youth's attitudes and blaming are predictors of perceived burden, values, and empathy, or that all examined variables are associated in a non-directional way (e.g., by all being explained by a third variable). Thus, the present study reflects only a first explorative step, and more follow-up research is needed in a representative longitudinal sample.

Third, individuals' responses are influenced by the environments they are in–a consideration especially relevant for the present study, as we examined context-specific experiences and attitudes. The data collection for the study ran for several months in the summer and fall of 2020, and the COVID-19 situation changed across that period, as well as before and after it. As a result, the number of cases and the strictness of the national approach was not the same when different individuals filled in the questionnaire. In the future, it would be useful to include a measure of the COVID-19 situation at the exact moment that each person participated in the data collection, as a better proxy of the environment in which young adults were embedded.

Finally, for all effects, even those highly significant, it is important to note that effect sizes were small [53], and that explained variance in the outcome variables was limited. Other, perhaps more situational factors such as the strictness of the government approach at each moment may have been at play that contribute to understanding individual differences in youth's attitudes towards the government approach and blaming of certain groups. It is likely that these fluctuating situational and other contextual factors may have played a larger role in a situation that is highly non-normative than stable, more distant factors such as personal values and traits.

## Conclusion

The present study examined individual differences in youth's attitudes towards their government and other social groups during times of COVID-19. We found evidence that a lower perceived burden due to COVID-19, higher collectivistic and lower individualistic values, and a higher level of empathy were each related to more positive attitudes towards the government approach and less blaming of certain groups, and that these associations were invariant across countries. Moreover, burden was indirectly related to less positive attitudes via more social identification. Burden, collectivistic values, and empathy were indirectly related to more blaming. The findings demonstrate support for social identity theory, which posits that identification with others may be a mechanism for individual attitudes and behavior. The findings provide a foundation for further research that can examine the specific groups that young adults identify with, and the specific groups that they might blame in future pandemics and when they experience pandemic-related government policies.

## Supporting information

**S1 Table. Descriptive statistics and demographics per country.**
(DOCX)

**S2 Table. Newly designed perceived COVID-19 burden measure used in the present study.**
(DOCX)

**S3 Table. Constrained multigroup mediation model with attitude towards government approach and blaming of certain groups regressed on COVID-19 burden, individualistic values, collectivistic values, and empathy, mediated by social identification; grouping on cumulative COVID-19 cases per 100,000 inhabitants.**
(DOCX)

**S4 Table. Constrained multigroup mediation model with attitude towards government approach and blaming of certain groups regressed on COVID-19 burden, individualistic values, collectivistic values, and empathy, mediated by social identification; grouping on strictness of the national COVID-19 approach.**
(DOCX)

**S5 Table. Constrained multigroup mediation model with attitude towards government approach and blaming of certain groups regressed on COVID-19 burden, individualistic values, collectivistic values, and empathy, mediated by social identification; grouping on national level individualistic values.**
(DOCX)

## Acknowledgments

We want to thank all international collaborators for their help in collecting the data. We additionally thank all participants who contributed their unique experiences during the COVID-19 pandemic to this project.

## Author Contributions

**Conceptualization:** Elisabeth L. De Moor, Christian Berger, Alexia Carrizales, Claire F. Garandeau, Maria Gerbino, Skyler T. Hawk, Goda Kaniušonytė, Asiye Kumru, Elisabeth Malonda, Anna Rovella, Yuh-Ling Shen, Laura K. Taylor, Maarten van Zalk, Susan Branje, Gustavo Carlo, Laura Padilla Walker, Jolien Van der Graaff.

**Data curation:** Ting-Yu Cheng, Jenna E. Spitzer, Jolien Van der Graaff.

**Formal analysis:** Elisabeth L. De Moor.

**Funding acquisition:** Jolien Van der Graaff.

**Methodology:** Ting-Yu Cheng, Jenna E. Spitzer, Jolien Van der Graaff.

**Supervision:** Jolien Van der Graaff.

**Writing – original draft:** Elisabeth L. De Moor, Ting-Yu Cheng, Jenna E. Spitzer, Susan Branje, Gustavo Carlo, Laura Padilla Walker, Jolien Van der Graaff.

**Writing – review & editing:** Elisabeth L. De Moor, Ting-Yu Cheng, Jenna E. Spitzer, Christian Berger, Alexia Carrizales, Claire F. Garandeau, Maria Gerbino, Skyler T. Hawk, Goda Kaniušonytė, Asiye Kumru, Elisabeth Malonda, Anna Rovella, Yuh-Ling Shen, Laura K. Taylor, Maarten van Zalk, Susan Branje, Gustavo Carlo, Laura Padilla Walker, Jolien Van der Graaff.

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
