## [Decision Letter · Decision Letter 0]

13 Jul 2022

PONE-D-22-10911What should I do and who's to blame? A cross-national study on youth's attitudes and beliefs in times of COVID-19PLOS ONE

Dear Dr. De Moor,

Thank you for submitting your manuscript to PLOS ONE. After careful consideration, we feel that it has merit but does not fully meet PLOS ONE’s publication criteria as it currently stands. Therefore, we invite you to submit a revised version of the manuscript that addresses the points raised during the review process.

As you can see, the reviews are in general favourable. However, both reviewers suggest some changes to your manuscript. After considering the reviews and reading the paper myself, I offer a number of points to address in a potential revision, as well.

1. Please explain how the scales were translated into the language of the participants’ country? It is also unclear whether the measures of culture orientation, empathy and social identification have been validated for use in the 14 study countries.

2. Although individualism/collectivism is perhaps the most commonly applied construct in explaining cultural effects, other dimensions of culture identified by Hofstede, e.g. power distance, masculinity-femininity uncertainty avoidance, long- vs. short-term orientation and indulgence-restraint, may also be relevant. It would be useful if the authors acknowledge the need for further cross-cultural research on this topic.

We look forward to receiving your revised manuscript.

Kind regards,

Helena R. Slobodskaya, M.D., Ph.D., D.Sc.

Academic Editor

PLOS ONE

Journal Requirements:

2.  Peer review at PLOS ONE is not double-blinded (https://journals.plos.org/plosone/s/editorial-and-peer-review-process). For this reason, authors should include in the revised manuscript all the information removed for blind review.

“Elisabeth de Moor and Susan Branje were supported by a grant from the European Research Council (ERC-CoG INTRANSITION-773023). Jolien Van der Graaff received a grant from the COVID-19 fund of the Faculty of Social Sciences of Utrecht University.”

“JvdG received a grant from the Faculty of Social Sciences of Utrecht University to fund data collection (no grant number available). URL: https://www.uu.nl/en/organisation/faculty-of-social-and-behavioural-sciences

ELdM and SB were supported by a grant from the European Research Council (ERC-CoG INTRANSITION-773023). URL: https://erc.europa.eu/

5. We note that you have referenced (Unpublished doctoral dissertation, University of Utah and Unpublished manuscript ) which has currently not yet been accepted for publication. Please remove this from your References and amend this to state in the body of your manuscript: (ie “Bewick et al. [Unpublished]”) as detailed online in our guide for authors

Reviewers' comments:

Reviewer's Responses to Questions

**Comments to the Author**

1. Is the manuscript technically sound, and do the data support the conclusions?

Reviewer #1: Yes

Reviewer #2: Partly

2. Has the statistical analysis been performed appropriately and rigorously? 

Reviewer #1: Yes

Reviewer #2: N/A

3. Have the authors made all data underlying the findings in their manuscript fully available?

Reviewer #1: No

Reviewer #2: No

4. Is the manuscript presented in an intelligible fashion and written in standard English?

Reviewer #1: Yes

Reviewer #2: Yes

5. Review Comments to the Author

Reviewer #1: the work is interesting and well written. Some aspects that can improve the work are emphasized.

An in-depth study of country-specific literature in the construction of the theoretical part with respect to the topics under investigation, including the part on the restrictions of different governments.

It would be important to test the power of the overall sample and individual countries.

In addition, the application implications of the work should be developed

Reviewer #2: This article discusses a relevant topic in the current context: what are the determinants (or some of them) of young people's attitudes to public health measures put in place during the pandemic. The topic is of global interest and appropriate for a multidisciplinary journal such as PLOS ONE.

ABSTRACT:

The authors identify their research object rather clearly in the abstract, although I suggest describing the results of their model in greater detail and precision. Also, '(...) blaming certain groups' seems too vague phrasing to me. The authors need to be more specific about which "groups" they refer to in the abstract and the text in general (and in the model). Ideally, it would also be helpful to discuss (along the manuscript) what is the meaning of blaming these groups in the youth context of these countries. I also recommend adding a sentence to put the theory of "social identification" in context, so that it is made more understandable in an interdisciplinary context.

Keywords should be words, not sentences.

INTRODUCTION:

It seems to me that the authors in this section offer an adequate amount of information, but that it should be better organized. First, the authors seem to arrive at Figure 1 too quickly. This conceptual model should be better justified (the identification of these factors and the mediating mechanisms) and it would also help to make the hypotheses derived or underlying them more explicit. Some hypotheses are made but not formally highlighted.

I would thus reorganize the section so that the reader is first explained the rationale for the identification of the determinants and hypotheses. After that, I would offer the conceptual model (which could possibly also be moved to the methodological part as an analysis model).

One recent reference that could help to add in this section on young people:

Vacchiano, M., (2022). How the First COVID-19 Lockdown Worsened Younger Generations' Mental Health: Insights from Network Theory. Sociological Research Online, https://doi.org/10.1177/13607804221084723

METHODS:

The methodological part is sufficiently detailed. One fundamental doubt:

I am not sure it can be said 100% that a level 2 of 14 units does not need a multilevel analysis. I know there is debate on the issue, but as far as I know the best way to measure effects at the country level is through multilevel analysis. According to Austin (2010) even models with 10 level-2 clusters could produce reliable statistical inference (see also Bryan and Jenkins 2016 on this controversial issue).

I have no expertise to evaluate the strategy that the authors have set up to test the effect of countries. I am not saying it is wrong, I simply don't know it and fully understand it. Personally, I think a multilevel analysis would be sounder.

RESULTS, DISCUSSION, LIMITATIONS:

On the rest of the manuscript, I have little to add. It seems to me that the paper brings out some interesting results and that the authors identify well the limitations of their work and the data.

In general, the paper is publishable with appropriate modifications. First, (1) reorganizing the introductory section and making explicit in a formal way the hypotheses (and so justifying the conceptual model). Second, the (2) authors need to better justify why multilevel analysis is not performed (here a paper https://doi.org/10.1093/esr/jcv059 that may help). Multilevel analysis could still be necessary in this paper. Also, in the discussion the differences between countries are not mentioned at all, which casts more doubt on the adequacy of the strategy pursued by the authors to capture country effects. I am sure the authors can offer more interesting arguments on this country differences.

All this would certainly benefit the quality and impact of this article.

6. PLOS authors have the option to publish the peer review history of their article (what does this mean?). If published, this will include your full peer review and any attached files.

Reviewer #1: No

Reviewer #2: No

---

## [Author Response · Author response to Decision Letter 0]

7 Oct 2022

Please see uploaded Response to Reviewers letter

---

## [Decision Letter · Decision Letter 1]

7 Dec 2022

What should I do and who's to blame? A cross-national study on youth's attitudes and beliefs in times of COVID-19

PONE-D-22-10911R1

Dear Dr. De Moor,

We’re pleased to inform you that your manuscript has been judged scientifically suitable for publication and will be formally accepted for publication once it meets all outstanding technical requirements.

Kind regards,

Helena R. Slobodskaya, M.D., Ph.D., D.Sc.

Academic Editor

PLOS ONE

Reviewers' comments:

Reviewer's Responses to Questions

**Comments to the Author**

1. If the authors have adequately addressed your comments raised in a previous round of review and you feel that this manuscript is now acceptable for publication, you may indicate that here to bypass the “Comments to the Author” section, enter your conflict of interest statement in the “Confidential to Editor” section, and submit your "Accept" recommendation.

Reviewer #2: All comments have been addressed

2. Is the manuscript technically sound, and do the data support the conclusions?

Reviewer #2: Partly

3. Has the statistical analysis been performed appropriately and rigorously? 

Reviewer #2: N/A

4. Have the authors made all data underlying the findings in their manuscript fully available?

Reviewer #2: Yes

5. Is the manuscript presented in an intelligible fashion and written in standard English?

Reviewer #2: Yes

6. Review Comments to the Author

Reviewer #2: The authors have substantially improved the original version of the manuscript and responded to my comments, resolving where possible the problems highlighted.

7. PLOS authors have the option to publish the peer review history of their article (what does this mean?). If published, this will include your full peer review and any attached files.

Reviewer #2: No

---

## [Editor Report · Acceptance letter]

12 Dec 2022

PONE-D-22-10911R1 

What Should I do and Who's to Blame? A Cross-National Study on Youth’s Attitudes and Beliefs in Times of COVID-19 

Dear Dr. De Moor:

I'm pleased to inform you that your manuscript has been deemed suitable for publication in PLOS ONE. Congratulations! Your manuscript is now with our production department. 

Kind regards, 

on behalf of

Dr. Helena R. Slobodskaya 

Academic Editor

PLOS ONE